# Association between air quality index and effects on emergency department visits for acute respiratory and cardiovascular diseases

Panumas Surit[1], Wachira Wongtanasarasin[1], Chiraphat Boonnag[2,3], Borwon Wittayachamnankul[1] *

1 Department of Emergency Medicine, Faculty of Medicine, Chiang Mai University, Chiang Mai, Thailand,
2 Department of Biochemistry, Faculty of Medicine, Chiang Mai University, Chiang Mai, Thailand,
3 Biomedical Informatics Center, Faculty of Medicine, Chiang Mai University, Chiang Mai, Thailand

* borwon.witt@cmu.ac.th

## Abstract

### Background and objective

Several studies suggest that air pollution, particularly PM2.5, increases morbidity and mortality, Emergency Department (ED) visits, and hospitalizations for acute respiratory and cardiovascular diseases. However, no prior study in Southeastern Asia (SEA) has examined the effects of air pollutants on ED visits and health outcomes. This study focused on the association of the Air Quality Index (AQI) of $PM_{2.5}$ and other pollutants' effects on ED visits, hospitalization, and unexpected deaths due to acute respiratory disease, acute coronary syndrome (ACS), acute heart failure (AHF), and stroke.

### Methods

We conducted a retrospective study with daily data from ED visits between 2018 and 2019 at Maharaj Nakorn Chiang Mai Hospital, Chiang Mai, Thailand. The AQI of air pollution data was collected from outdoor air quality from the Smoke Haze Integrated Research Unit and the Air Quality Index Visual Map. A distributed lag, non-linear and quasi-Poisson models were used to explore the relationship between air quality parameters and ED visits for each disease.

### Results

3,540 ED visits were recorded during the study period. The mean daily AQI of $PM_{2.5}$ was 89.0 ± 40.2. We observed associations between AQI of $PM_{2.5}$ and the ED visits due to ACS on the following day (RR = 1.023, 95% confidence interval [CI]: 1.002–1.044) and two days after exposure (RR = 1.026, 95% CI: 1.005–1.047). Also, subgroup analysis revealed the association between AQI of $PM_{2.5}$ and the ED visits due to pneumonia on the current day (RR = 1.071, 95% CI: 1.025–1.118) and on the following day after exposure (RR = 1.024, 95% CI: 1.003–1.046). AQI of $PM_{2.5}$ associated with increased mortality resulted from ACS on lag day 3 (OR = 1.36, 95% CI: 1.08–1.73). The AQI of $PM_{10}$ is also associated with increased ED visits due to COPD/asthma and increased hospitalization in AHF. In addition,

**Data Availability Statement:** All relevant data are within the paper and its Supporting Information files.

**Funding:** Faculty of Medicine, Chiang Mai University funded this research (Fund No. 089-2563). The funders had no role in study design, data collection and analysis, decision to publish, or preparation of the manuscript.

**Competing interests:** The authors have declared that no competing interests exist.

the AQI of $O_3$ and AQI of $NO_2$ is associated with increased ICU admissions and mortality in AHF.

## Conclusion

Short-term PM2.5 exposure escalates ED visits for ACS and pneumonia. PM10's AQI associates with COPD/asthma ED visits and AHF hospitalizations. AQI of $O_3$ and $NO_2$'s link to increased ICU admissions and AHF mortality. Urgent action against air pollution is vital to safeguard public health.

## Introduction

Air pollution is a major public health concern worldwide and represents one of the largest environmental problems [1]. Outdoor air pollution is a complex mixture of components affecting human health, including airborne particulate matter (PM), pollutants ozone ($O_3$), nitrogen dioxide ($NO_2$), and sulfur dioxide ($SO_2$) [2]. The most health-damaging PMs are those with a diameter of less than 10 μm ($PM_{10}$) and 2.5 μm ($PM_{2.5}$), which can penetrate and lodge deep inside the lungs. The principal air pollutant of $PM_{2.5}$ shows the greatest threat to global public health [2, 3]. Both short- and long-term exposure to $PM_{2.5}$ has been associated with health impacts in multiple organ systems via many pathways with the role of oxidative stress in PM-mediated effects, systemic vascular dysfunction, and cardiovascular modeling with air pollution, and autonomic dysfunction and activation of central nervous system pathways. Exposure to $PM_{2.5}$ also leads to respiratory problems and the development of atherosclerosis which increases the risk for coronary artery disease and cerebrovascular disease [3]. Several studies have demonstrated the effects of $PM_{2.5}$ on increased morbidity and mortality, emergency visit, and hospitalization for acute respiratory problems, including pneumonia, chronic obstructive pulmonary disease (COPD) and asthma, acute coronary syndrome (ACS), acute heart failure (AHF), and stroke [2–8].

According to the World Health Organization (WHO) global assessment of ambient air pollution exposure and the resulting burden of disease in 2016, the Eastern Mediterranean, South-East Asian (SEA), and Western Pacific Regions had some of the highest exposures to air pollution [9–11]. Thailand is a SEA country with exposure to high annual mean WHO Air Quality Guidelines (AQG) [11, 12]. The area with the highest air pollution concentrations is northern Thailand, specifically from January to May. The pollution originates from human activity and wildfires; [12] however, few study in SEA mentions the effects of $PM_{2.5}$ on health impacts in ED visits. Therefore, the primary objective of this study was to determine the association between the increase of fine particulate matter ($PM_{2.5}$) and other pollutants, and its effect on acute respiratory disease, ACS, AHF, and stroke in the ED visits, ED disposition, and in-hospital mortality.

## Material and methods

### Data collection

Data was retrospectively collected between April 2018 and March 2019 at Maharaj Nakorn Chiang Mai Hospital, a tertiary care and university hospital. Inclusion criteria were patients who visited the ED older than 18 with a current address in Chiang Mai. The exclusion criteria were patients referred from other hospitals and trauma patients. The study protocol was

approved by the Research Ethics Committee, Faculty of Medicine, Chiang Mai University (Permit No. EXEMPTION-6698/2562). The institutional Ethics Committee waived the need for consent.

From the electronic medical record (EMR), we extracted patient data of ED patients with the principal diagnosis of acute respiratory disease, ACS, AHF, and stroke, using only the International Classification of Diseases (ICD) 10th revision codes of J00-J99, I20-I25, I50, and I60-I69, respectively. The principal diagnosis was recorded in EMR by ED physicians. Data collection derived from the EMR included age, gender, visit date, diagnosis, Emergency Department disposition, and in-hospital mortality.

We examined the daily outdoor Air Quality Index (AQI) for pollutants including PM2.5, PM10, ozone ($O_3$), nitrogen dioxide ($NO_2$), and sulfur dioxide ($SO_2$). The AQI serves as a standardized metric for assessing and communicating air pollution levels. It is calculated by converting the concentrations of these pollutants into sub-index values using established equations or algorithms (S1 File). These sub-index values are then combined to derive the overall AQI value, indicating the air quality level daily. However, we utilized these individual AQI values (sub-index value) collectively to conduct our analysis. These were collected from the Smoke Haze Integrated Research Unit (SHIRU) and the Air Pollution in Chiang Mai: Real-time Air Quality Index Visual Map website [13]. Twelve stations are on the map in the Muang Chiang Mai District of Chiang Mai Province in Thailand, managed by The World AQI project team. The outcome of this study has measured the effects of air pollutions on ED visits, hospitalization, Intensive Care Unit (ICU) admissions, and mortality from acute respiratory disease, ACS, AHF, and stroke.

## Data analysis

The results were presented as medians and interquartile ranges for non-normally distributed variables and frequency with percentages for categorical variables. The primary exposure variable was the AQI of $PM_{2.5}$. Considering the nonlinear exposure-lag-response relationship between exposure to air pollution and health effects, an additional dimension, temporal dependency of exposure and outcome, was required to characterize and control the model [14–17]. Distributed lag non-linear model (DLNM) is a model in which the relationship between air quality parameters and ED visits is described in the usual predictor and the additional dimension of time lags. The model is defined by the following formula:

$$\log(E(Y_t)) = \alpha + ns(RH_t, 3) + ns(Temperature, 3) + ns(Focused\_AP_t, 3)$$

$$+ \sum_{i=0}^{q} \beta_i (Focused\_AP)_{t-i} + \varepsilon_t$$

Where $Focused\_AP \in \{AQI\ of\ PM_{2.5}, AQI\ of\ PM_{10}\}$ and

$Other\_AP \in \{AQI\ of\ PM_{2.5}, AQI\ of\ PM_{10}, AQI\ of\ O_3, AQI\ of\ N_2\} - \{Focused\_AP\}$

$E(Y_t)$ is the expected number of ED visits on day t, and $\alpha$ is the model intercept. Focused_AP$_{t-q}$ represented the focused daily average AQI of $PM_{2.5}$ or $PM_{10}$ (based on which pollutant we focused) q days before the ED visit. The model was adjusted for other environmental confounding variables using a natural cubic spline with three df for relative humidity ($RH_t$), temperature and AQI of other air pollutants (Other_AP$_t$; $PM_{10}$ (or $PM_{2.5}$ for model that focused on $PM_{10}$), $O_3$, and $NO_2$). We investigated the lag structure of AQI of $PM_{2.5}$ effects on ED visits using a polynomial function with seven days from lag day 0 to lag day 6. The outcome

of the DLNM was the number of ED visits for each disease. A multivariable logistic regression model was employed to explore the independent AQI of each pollutant ($PM_{2.5}$, $PM_{10}$, $O_3$, $NO_2$, and $SO_2$) predictors of hospital admission, ICU admission, and in-hospital death. All statistical analyses were performed using R (version 4.0.0). Two-tailed $p < 0.05$ was considered statistically significant. DLNM package version 2.4.2 was used for model development.

## Results

Of the total, 3,540 ED visits were recorded for acute respiratory diseases, ACS, AHF, and stroke, from April 2018 to March 2019, with respiratory disease accounting for the largest proportion (55.7%). Most patients were over 65 (46.3%) and male (51.5%). The highest percentage of daily ED visits for cause-specific respiratory diseases were upper respiratory tract infections (URTI) (28.5%), followed by pneumonia (27.3%), chronic obstructive pulmonary disease (COPD), and asthma (26%), respectively. 1,664 out of 3,540 (47%) were admitted to the hospital, and 16.6% were admitted to the ICU. Patients diagnosed with ACS had the highest admission rate (80.6%). **Table 1** describes the characteristics of patients presented to the ED during the study period. The overall median daily AQI of $PM_{2.5}$ was 75.0 (Interquartile range; IQR 63–102), and the mean daily AQI of $PM_{2.5}$ was 89.0 ± 40.2 above from "Good level" (0–50) AQI recommendations. The median daily AQI of $PM_{10}$, $O_3$, $NO_2$, and $SO_2$ were 39 (IQR 32–53), 19 (IQR 15–24), 8 (IQR 6–12), and 0, respectively, as shown in **Fig 1** **and S1 Table in S1 File**. The onset of the summer season heralded the culmination of peak levels in the AQI for all air pollutants. Specifically, this phenomenon was observed from April 2018 to March 2019, as demonstrated in **S2 Fig in S1 File**.

**Table 1. Descriptive summary for the study period (April 2018 to March 2019), emergency department visits, ICU admission, and in-hospital deaths for acute respiratory diseases, acute coronary syndrome, acute heart failure, and stroke.**

| Characteristics | Total | Acute Respiratory Disease (ICD10 code: J00-J99) | Acute Coronary Syndrome (ICD10 code: I20-I25) | Acute Heart Failure (ICD10 code: I50) | Stroke (ICD10 code: I60-69) |
|---|---|---|---|---|---|
| | N (%) | N (%) | N (%) | N (%) | N (%) |
| ED visit | 3540 (100) | 1973 (55.7) | 470 (13.3) | 365 (10.3) | 732 (20.7) |
| Median age–year (IQR) | 63 (29) | 59 (44) | 65 (15.8) | 69 (20) | 63 (21) |
| Age (year) * | | | | | |
| 18–34 | 637 (18.0) | 596 (30.2) | 1 (0.2) | 8 (2.2) | 32 (4.4) |
| 35–44 | 207 (5.8) | 135 (6.8) | 18 (3.8) | 14 (3.8) | 40 (5.5) |
| 45–54 | 349 (9.9) | 141 (7.2) | 59 (12.6) | 35 (9.6) | 114 (15.6) |
| 55–64 | 705 (20.0) | 291 (14.8) | 140 (29.8) | 67 (18.4) | 207 (28.3) |
| ≥ 65 | 1638 (46.3) | 810 (41.0) | 252 (53.6) | 241 (66.0) | 335 (45.8) |
| Gender | | | | | |
| Male | 1824 (51.5) | 988 (50.1) | 293 (62.3) | 158 (43.3) | 385 (52.6) |
| Female | 1716 (48.5) | 985 (49.9) | 177 (37.7) | 207 (56.7) | 347 (47.4) |
| Disposition | | | | | |
| Discharge | 1876 (53.0) | 1414 (71.7) | 91 (19.4) | 156 (42.7) | 215 (29.4) |
| Total Admission | 1664 (47.0) | 559 (28.3) | 379 (80.6) | 209 (57.3) | 517 (70.6) |
| ICU Admission | 276 (16.6) | 22 (3.9) | 218 (57.5) | 20 (9.6) | 16 (3.09) |
| In-hospital death | 153 (9.2) | 75 (13.4) | 27 (7.1) | 19 (9.1) | 32 (6.2) |

* Note: Age (N/A = 4; ICD10 I60-I69 = 4

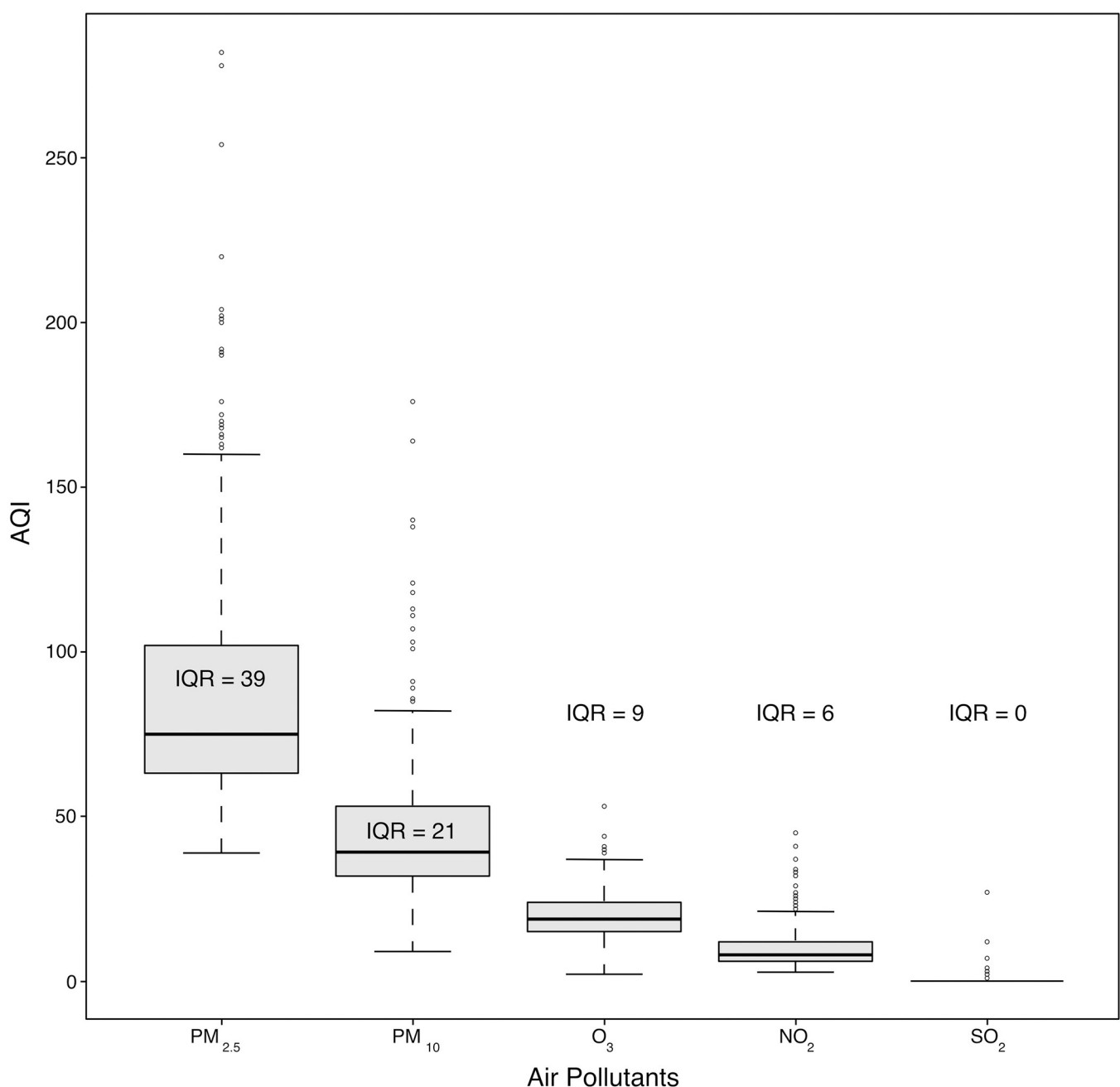

**Fig 1. Box plot of daily AQI of each pollutants for the study period (April 2018-March 2019).**

**Fig 2** shows the association between the adjusted lag-effect of AQI of $PM_{2.5} > 50$ on ED visits to acute respiratory disease, ACS, AHF, and stroke. We observed statistically significant associations between ED visits due to ACS and AQI of $PM_{2.5}$ on the following days (lag day 1) (RR = 1.023; 95% CI: 1.002–1.044), two days after exposure (lag day 2) (RR = 1.026; 95% CI: 1.005–1.047) and returned to normal three days after exposure to $PM_{2.5}$ (lag day 3). No statistically significant associations were found between AQI of $PM_{2.5}$ and ED visits due to acute respiratory disease, AHF, and stroke for any lag day.

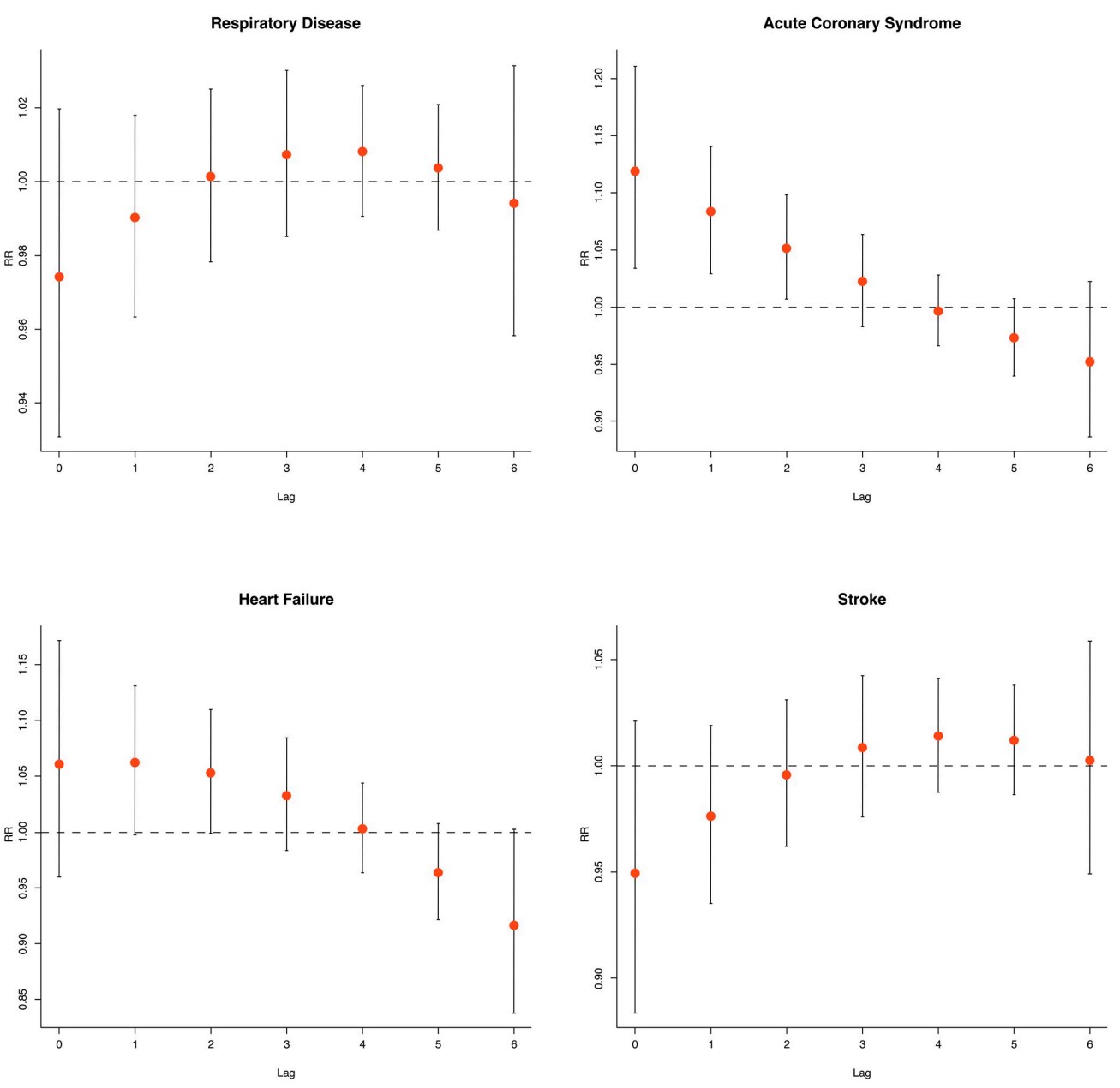

**Fig 2. Relative risk of the adjusted lag-effect between AQI of PM$_{2.5}$ and ED visits of respiratory disease, acute coronary syndrome, acute heart failure, and stroke (Reference AQI of PM$_{2.5}$ = 50).**

Analysis of the effect of AQI of PM$_{2.5}$ on cause-specific respiratory diseases, including URTI, pneumonia, COPD, and asthma (**Fig 3**) found statistically significant associations in ED visits of pneumonia on the current day (lag day 0) (RR = 1.071; 95% CI: 1.025–1.118), lag day 1 (RR = 1.024; 95% CI: 1.003–1.046) then reverted to normal levels at two days after exposure (lag day 2), with the effects rebounding four days after exposure (lag day 4) (RR = 0.970; 95% CI: 0.948–0.993) and five days after exposure (lag day 5) (RR = 0.979; 95% CI: 0.961–

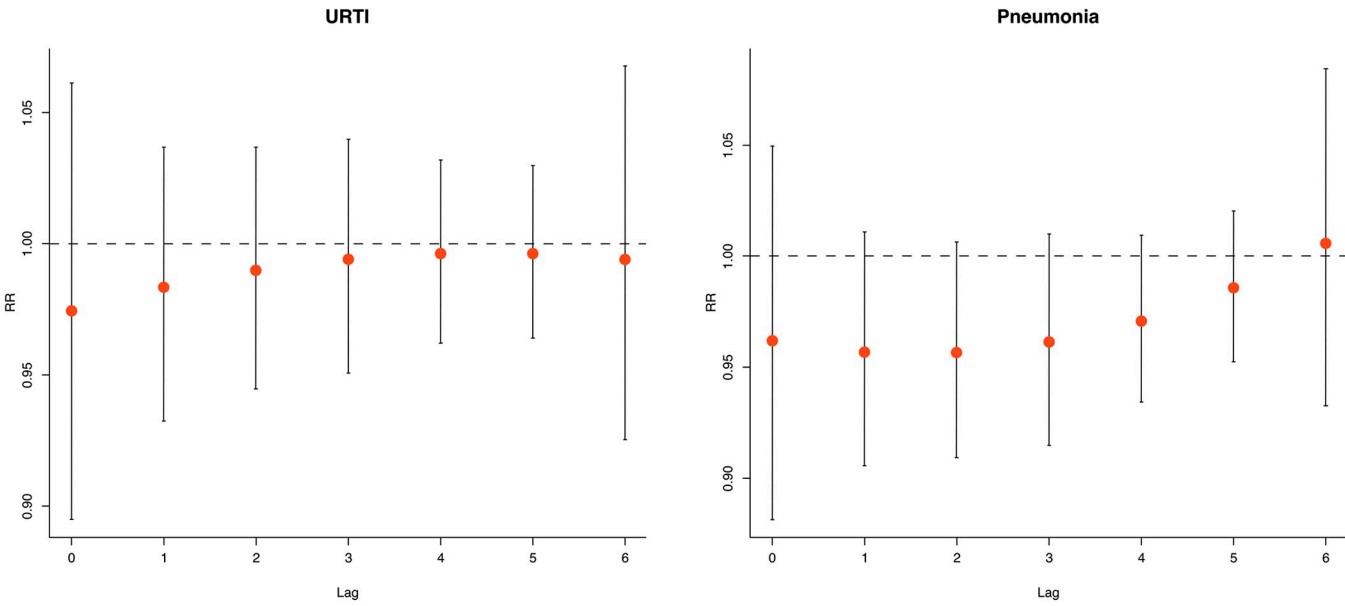

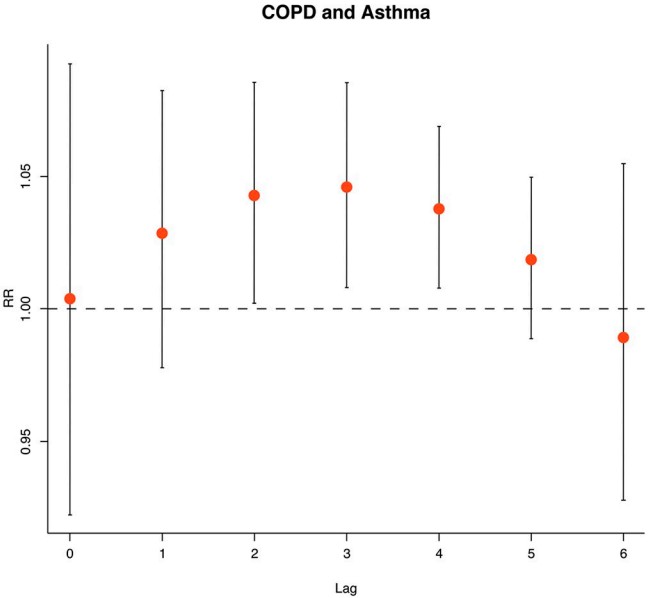

**Fig 3. Relative risk of the adjusted lag-effect between PM$_{2.5}$ and ED visits of cause-specific of respiratory disease; URTI, pneumonia, COPD, and asthma.** (Reference AQI of PM$_{2.5}$ = 50).

0.998). The largest effect was observed on the current day (lag day 0). Moreover, we investigated the effects of AQI of other air pollutants, such as PM$_{10}$, O$_3$, and NO$_2$, on ED visits and hospitalizations. Above the moderate level of AQI of PM$_{10}$ (>120) were associated with increased ED visits due to COPD and asthma on lag day three (RR = 1.159; 95% CI: 1.007–1.1335) and lag day four (RR = 1.138; 95% CI: 1.009–1.284) **(S3 Fig in S1 File).** For O$_3$ and NO$_2$, no conclusions could be drawn since their AQI was within the standard references across the study period.

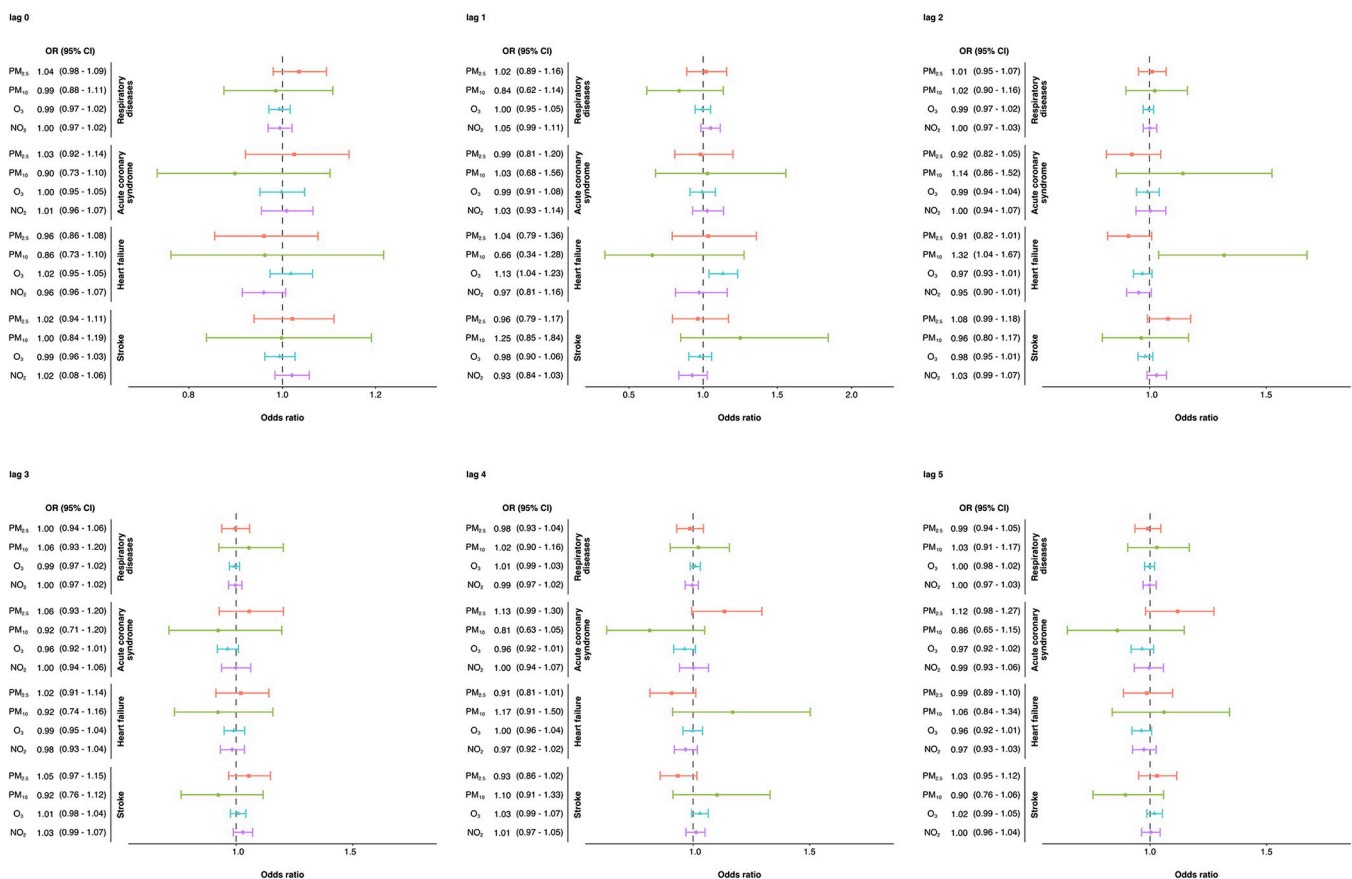

**Fig 4. Excess risk (95% confidence intervals) of association between AQI of each air pollutants (PM2.5, PM10, O3, and NO2) and hospitalization for serious specific disease at lag day 0–5.**

Fig 4 demonstrates the association between air pollutants and hospitalization for specific diseases on any lag day. We observed statistically significant associations between the hospitalization and AHF with every 10 increase of AQI in $PM_{10}$ on lag day two (OR = 1.32; 95% CI: 1.04–1.67). However, the statistical results show no significant associations between air pollutants and other diseases. Additionally, this study found statistically significant air pollutants increase the risk effect of ICU admissions and mortality on any lag day, as shown in **S4 and S5 Figs in S1 File**. Every 10 AQI increase of $NO_2$ on the previous two days was associated with an increase in ICU admission for AHF (OR = 1.13; 95% CI: 1.02–1.26). An AQI of $PM_{2.5}$ was associated with increased ACS mortality three days after exposure (OR = 1.36; 95% CI: 1.08–1.73). Furthermore, AQI of $O_3$ was associated with increased AHF mortality on the current day (OR = 1.16; 95% CI: 1.04–1.29), the following day (OR = 1.13; 95% CI: 1.04–1.23), and the next two days after exposure (OR = 1.17; 95% CI: 1.05–1.30).

## Discussion

This study found that the AQI of PM2.5 was associated with increased ED visits due to ACS and pneumonia (lag days 1 and 2). A significant relationship was observed between PM2.5 AQI and pneumonia on the current day and the following day after exposure. Consistent with previous studies, ED visits due to acute respiratory diseases were associated with an increased $PM_{2.5}$ concentration on the same day [4, 6, 18–20]. Interestingly, the previous report

investigated the effects of $PM_{2.5}$ on daily mortality and hospitalization in cardiovascular and respiratory diseases, which found $PM_{2.5}$ affecting COPD and community-acquired pneumonia [21]. Our findings correlated with the previous study showing that $PM_{2.5}$ is associated with increased mortality of ACS patients three days after exposure [19]. Additionally, we found that the effect of $PM_{2.5}$ pollution on respiratory disease hospitalization was significantly greater in males over 65 years. Compared with previous studies, the subgroup analysis results suggested that older people might be more susceptible to $PM_{2.5}$ exposure. They may have a weaker immune system and a higher prevalence of chronic respiratory diseases [6, 21].

$PM_{2.5}$ was hypothesized to have the most effects on the respiratory and cardiovascular systems. PM-mediated effects directly affects the respiratory system via the oxidative stress generation through inhaled $PM_{2.5}$ depositing deep within pulmonary tissues, interacting with local cells, and modifying endogenous structures. Also, PM-mediated effects influence the development of atherosclerosis, resulting in an increased risk of coronary artery disease [2–4]. The results of this study indicate that the short-term effect of ambient $PM_{2.5}$ exposure is associated with increased daily ED visits with ACS. The number of ACS patients increased following the rising of $PM_{2.5}$ AQI by one to two days. Increased AQI of PM2.5 did not affect the number of ED visits in lag days regarding AHF and stroke. Previous literature demonstrated that most $PM_{2.5}$ effects were delayed, ranging from seven days or more after exposure to $PM_{2.5}$; moreover, stroke was associated with long exposure to air pollution [5, 8, 22, 23].

Other air pollutants also demonstrated some effects on ED patients. Interestingly, $PM_{10}$ is another significant air pollutant that can precipitate emergency illnesses, especially cardiovascular and respiratory diseases; however, studies exploring the effect of $PM_{10}$ concentration and specific conditions are limited [3, 24]. We found that AQI of $PM_{10}$ was associated with an increased risk of ED visits due to COPD and asthma. Furthermore, our study revealed the association between AQI of $PM_{10}$ and an increased risk of hospitalization due to AHF. Correlated to the previous research, $PM_{10}$ concentrations between 50–200 $\mu g/m^3$ were an isolated risk factor for hospitalization in AHF patients in Saharan desert dust [25]. Also, $NO_2$ and $O_3$ also took part in the pathogenesis of inflammation, oxidative stress, and autonomic abnormality, resulting in an increased risk of unfavorable outcomes in AHF patients [3, 26, 27].

This is the first study in SEA studying the effects of AQI of $PM_{2.5}$ and $PM_{10}$ on ED visits in a place where the daily AQI of $PM_{2.5}$ and $PM_{10}$ were higher than the standard level (US AQI). Also, this study demonstrated the possible health effects of exposure to air pollution, especially the effects of $PM_{2.5}$ on the cardiovascular and respiratory systems, as mentioned. This is important for both healthcare providers and public health authorities and could contribute to health promotion for the general public by increasing awareness of the effects of air pollution and preventative measures. Also, to raise government awareness of the gravity of air pollution issues to be aware of harmfulness and health hazards to the populations in this area. Finally, to help guide systems planning for health care professionals, especially those in the emergency services, to prepare for increases in ACS and respiratory disease visits to the ED in seasons where $PM_{2.5}$ concentrations are abnormally high, especially during winter (December to February), and peak again in forest fire season (April) **(S1 Fig in S1 File)**. A well-prepared protocol during the season, which has a high level of $PM_{2.5}$, is warranted to ensure patients and healthcare personnel that ED and hospital are administering this issue.

Some limitations of our study should be mentioned. First, it is a single-center study conducted only in one district; thus, the sample size studied in subgroup analyses was small, and only one year of data was included. Furthermore, patients referred from other hospitals were excluded from the study. As this was conducted in a tertiary teaching hospital, the admission criteria were strict, which may influence the number of admissions in disease subgroup analyses. Second, patient data obtained from the principal diagnosis made in the ED according to

ICD-10 did not include disease severity, risk factors, and initial management. Moreover, our data explored the effects of short-term exposure. Finally, we did not collect data on preventative measures taken by the sample population to lessen $PM_{2.5}$ exposure, such as using home air purifiers and wearing face masks. Further study should focus on two primary issues: multiple meteorological factors and the long-term effects of air pollutants.

## Conclusion

In summary, our study provides robust evidence demonstrating the association between short-term $PM_{2.5}$ pollution exposure and increased risks of ED visits for ACS and respiratory diseases, particularly pneumonia. Moreover, the AQI of PM10 shows associations with elevated ED visits related to COPD and asthma, as well as heightened hospitalization in AHF cases. Furthermore, the AQI of $O_3$ and $NO_2$ is linked to an increase in ICU admissions and mortality in AHF cases. These findings emphasize the need for effective measures to reduce $PM_{2.5}$ and other pollution levels and protect vulnerable populations from the adverse health impacts of air pollution.

## Supporting information

**S1 File. PM2.5 and ED visits supplementary.docx.**
(DOCX)

**S1 Data.**
(ZIP)

## Acknowledgments

The authors would like to thank the Smoke Haze Integrated Research Unit (SHIRU) for supporting the air pollution data. Also, our thanks go to Miss Rudklao Sairai and colleagues, the Research Unit of the Department of Emergency Medicine, Chiang Mai University, for supporting this study. We appreciate Dr. Nattikarn Atthapreyangkul for their English language editing and Dr. Phichayut Phinyo for their methodology and statistical advice. Finally, we would like to thank the Medical Records and Statistics Section of Maharaj Nakorn Chiang Mai Hospital for providing the data of patients in this study.

## Author Contributions

**Conceptualization:** Panumas Surit.

**Formal analysis:** Panumas Surit, Chiraphat Boonnag, Borwon Wittayachamnankul.

**Funding acquisition:** Wachira Wongtanasarasin, Borwon Wittayachamnankul.

**Investigation:** Panumas Surit, Wachira Wongtanasarasin, Chiraphat Boonnag.

**Methodology:** Panumas Surit, Chiraphat Boonnag.

**Project administration:** Panumas Surit, Wachira Wongtanasarasin.

**Resources:** Wachira Wongtanasarasin.

**Software:** Chiraphat Boonnag.

**Supervision:** Borwon Wittayachamnankul.

**Visualization:** Chiraphat Boonnag.

**Writing – original draft:** Panumas Surit, Wachira Wongtanasarasin, Borwon Wittayachamnankul.

**Writing – review & editing:** Panumas Surit, Wachira Wongtanasarasin, Chiraphat Boonnag, Borwon Wittayachamnankul.

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
