## [Decision Letter · Decision Letter 0]

15 Aug 2022

PONE-D-22-17550Association Between PM 2.5 and Effect of Emergency Department Visits for Acute Respiratory Disease, Acute Coronary Syndrome, Acute Heart Failure, and StrokePLOS ONE

Dear Dr. Wittayachmnankul,

Thank you for submitting your manuscript to PLOS ONE. After careful consideration, we feel that it has merit but does not fully meet PLOS ONE’s publication criteria as it currently stands. Therefore, we invite you to submit a revised version of the manuscript that addresses the points raised during the review process.

We look forward to receiving your revised manuscript.

Kind regards,

Oyelola A. Adegboye, PhD

Academic Editor

PLOS ONE

Journal Requirements:

   "Faculty of Medicine, Chiang Mai University funded this research (Fund No. 089-2563)."

  "The authors would like to thank the Faculty of Medicine, Chiang Mai University for funding this research and the Smoke Haze Integrated Research Unit (SHIRU) for supporting the air pollution data. Also, our thanks goes to Miss Rudklao Sairai and colleagues, the Research Unit of Department of Emergency Medicine, Chiang Mai University for providing support to this study. We appreciate Dr. Nattikarn Atthapreyangkul for English language editing and Dr.Phichayut Phinyo for methodology and statistical advice. Finally, we would like to thank the Medical Records and Statistics Section of Maharaj Nakorn Chiang Mai Hospital for providing the data of patients in this study."

 "Faculty of Medicine, Chiang Mai University funded this research (Fund No. 089-2563)."

6. Please upload a copy of Supporting Information Figures 1, 2, 3, 4 and 5 which you refer to in your text on page 10,6,7 .

7. Please upload a copy of Supporting Information Table 1 which you refer to in your text on page 6.

Reviewers' comments:

Reviewer's Responses to Questions

**Comments to the Author**

1. Is the manuscript technically sound, and do the data support the conclusions?

Reviewer #1: Yes

Reviewer #2: Partly

2. Has the statistical analysis been performed appropriately and rigorously? 

Reviewer #1: Yes

Reviewer #2: No

3. Have the authors made all data underlying the findings in their manuscript fully available?

Reviewer #1: Yes

Reviewer #2: No

4. Is the manuscript presented in an intelligible fashion and written in standard English?

Reviewer #1: Yes

Reviewer #2: No

5. Review Comments to the Author

Reviewer #1: Dear Dr. Oyelola A. Adegboye,

Thank you for inviting me to review this manuscript. Overall this study is well done and interesting. The paper could be accepted for publication into PLOS ONE after minor revisions.

Comments to the Authors:

Title P.1

The title of your study is “association between PM 2.5 and effect of emergency department visits for acute respiratory disease, acute coronary syndrome, acute heart failure, and stroke.” However, the authors didn’t only focus on PM2.5, but also on other air pollutants, including PM10, O3, NO2, SO2. A suitable study title should contain the critical keywords and predict the content of the research. Therefore, the keyword "PM 2.5" may be revised as "air pollutants."

Material and Methods P. 4

Your study investigated short-term air pollutant exposures and emergency department visits for acute respiratory diseases and cardiovascular diseases between 2018 and 2019. In general, the study periods of epidemiologic studies regarding the relationship between short-term exposure to particulate matter and diseases are more than one year or at least two years. It would be better if the time series analysis from 2018 to 2020 could be conducted.

Material and Methods P. 4

The sampling of this study is not representative of the exposure-outcome distributions in the overall population. Selection bias could have occurred when investigators use only one hospital data.

Material and Methods P. 4

The inclusion criteria of the study subjects are not clear. How did you define daily counts of emergency department visits for acute respiratory diseases and cardiovascular diseases? by ICD10 or ICD10 plus some kind of treatments from hospital electronic medical records?

Material and Methods P. 5

The analyses consider several important confounding variables, including age, gender, and season.

Discussion P.9

Although unmeasured confounding remained in this study, you provide important new evidence that short-term PM2.5, PM10 and other air pollutant exposures is significantly related to emergency department visits for acute respiratory diseases and cardiovascular diseases. Further research might be conducted to investigate the associations of long-term exposure to fine particulate matter and air pollutants with healthcare utilization.

Discussion P.10

According to previous studies, Chiang Mai has been facing severe problem of haze pollution over the past years. Did emergency department visits to treat respiratory diseases and cardiovascular diseases increase during haze episodes in your study? Had Maharaj Nakorn Chiang Mai Hospital been prepared to meet the surge in demand for medical treatment? It could be one of the confounding factors in your study and may produce spurious or distorted associations between exposure to air pollutants and increased emergency department visits. So, the authors may add methods to decrease the impact of confounding variables in the study design. This requires clarification.

Discussion P.10

Chiang Mai is surrounded by the mountain ranges of the Thai highlands and has a tropical savanna climate with hot, wet temperatures year-round. Many studies have shown the meteorological factors could impact the PM concentrations, including the dispersion, growth, chemical production, photolysis, and deposition of PM. Besides, the dominant meteorological factors for PM concentrations are closely related to geographical conditions. Also, weather conditions could trigger respiratory symptoms in patients with respiratory diseases. Thus, I suggest that the effects of PM concentrations on multiple meteorological factors, including cloud cover, temperature, wind, humidity, relative humidity, pressure, precipitation, rainfall, radiation and planetary boundary layer height might be included and carefully examined in the further study.

Conclusion P.10

The authors should be cautious in interpreting your results. Many studies focusing on PM2.5 and health outcomes may overstate conclusions of their findings.

Reviewer #2: This is an important research study into the acute effects of ambient air pollutants on ED visits for a variety of impotant outcomes. However, there are several limitations that either make me doubt the results and/or formations of the analysis that are unclear and incomplete. Major and minor comments are provided below.

Major Comments:

1. The exposure assessment metrics are not clearly stated in the abstract or in the methods. How were pollutants measured? What is the LOD and how were below LOD values addressed? So2 cannot be 0ug/m3, which makes me concerned about how below LOD values were handled.

2. It is unclear why PM2.5 was only modeled as 50ug/m3 as the reference group. What is the rationnale for this? Why not model a continuous measure? This analysis seems quite incomplete to me.

3. PM10 was substantially lower than PM2.5. This practically speaking cannot be possible because PM2.5 is a fraction of PM10. Authors need to explain this oddity. Was PM2.5 and PM10 somehow swapped? This is a major gap in the data that is not addressed by the authors.

4. There is a lot of ambiguities that need clearing up, which I believe is largely due to issues with the grammar. For instance, referring to exposure on the following day in the abstract, and repeating it later, makes it sound like exposure effects on the day after ED visit was associated with ED visits. This is inconsistent with how other day lags are talked about in the paper.

5. Overall, there needs to be much more extensive english language editing to fix the clarity of this manuscript.

6. Study figures, especially the last two, are of very poor quality and mostly unreadable. These must be more clear for the reader to understand the data.

Minor Comments:

1. The abstact background is lacking background information.

2. the abstract methods need to specify the study area (city, country) and the exposure assessment method.

3. What statistical package (and version number) did authors use for dlnm modeling? Need to specify these

6. PLOS authors have the option to publish the peer review history of their article (what does this mean?). If published, this will include your full peer review and any attached files.

Reviewer #1: **Yes: **Man-Ju Ting

Reviewer #2: **Yes: **Eric S Coker

---

## [Author Response · Author response to Decision Letter 0]

28 Oct 2022

Response to reviewer and editor

Answer: thank you for suggestion. We edit the revised manuscript to PLOS ONE's style requirements

 Answer: We retrospective collected data and the need for consent was waived by the ethics committee. We add this information in the revised manuscript.

 "Faculty of Medicine, Chiang Mai University funded this research (Fund No. 089-2563)."

Answer: Thanks, we change this in revised manuscript.

 "The authors would like to thank the Faculty of Medicine, Chiang Mai University for funding this research and the Smoke Haze Integrated Research Unit (SHIRU) for supporting the air pollution data. Also, our thanks goes to Miss Rudklao Sairai and colleagues, the Research Unit of Department of Emergency Medicine, Chiang Mai University for providing support to this study. We appreciate Dr. Nattikarn Atthapreyangkul for English language editing and Dr.Phichayut Phinyo for methodology and statistical advice. Finally, we would like to thank the Medical Records and Statistics Section of Maharaj Nakorn Chiang Mai Hospital for providing the data of patients in this study."

 "Faculty of Medicine, Chiang Mai University funded this research (Fund No. 089-2563)."

Answer: Thanks, we change this in revised manuscript.

Answer: We will update your Data Availability statement to reflect the information you provide in your cover letter.

6. Please upload a copy of Supporting Information Figures 1, 2, 3, 4 and 5 which you refer to in your text on page 10,6,7 .

Answer: Thank you, we just upload this information in revised manuscript

7. Please upload a copy of Supporting Information Table 1 which you refer to in your text on page 6.

Answer: Thank you, we just upload this information in revised manuscript

Reviewers' comments:

Reviewer's Responses to Questions

Comments to the Author

1. Is the manuscript technically sound, and do the data support the conclusions?

Reviewer #1: Yes

Reviewer #2: Partly

2. Has the statistical analysis been performed appropriately and rigorously?

Reviewer #1: Yes

Reviewer #2: No

3. Have the authors made all data underlying the findings in their manuscript fully available?

Reviewer #1: Yes

Reviewer #2: No

4. Is the manuscript presented in an intelligible fashion and written in standard English?

Reviewer #1: Yes

Reviewer #2: No

5. Review Comments to the Author

Reviewer #1: Dear Dr. Oyelola A. Adegboye,

Thank you for inviting me to review this manuscript. Overall this study is well done and interesting. The paper could be accepted for publication into PLOS ONE after minor revisions.

Comments to the Authors:

Title P.1

The title of your study is “association between PM 2.5 and effect of emergency department visits for acute respiratory disease, acute coronary syndrome, acute heart failure, and stroke.” However, the authors didn’t only focus on PM2.5, but also on other air pollutants, including PM10, O3, NO2, SO2. A suitable study title should contain the critical keywords and predict the content of the research. Therefore, the keyword "PM 2.5" may be revised as "air pollutants."

Answer: Thanks, we change this in revised manuscript.

Material and Methods P. 4

Your study investigated short-term air pollutant exposures and emergency department visits for acute respiratory diseases and cardiovascular diseases between 2018 and 2019. In general, the study periods of epidemiologic studies regarding the relationship between short-term exposure to particulate matter and diseases are more than one year or at least two years. It would be better if the time series analysis from 2018 to 2020 could be conducted.

Answer: Thank you for your suggestion, however, we have only one year of ER visit data we add this in limitation of revised manuscript.

Material and Methods P. 4

The sampling of this study is not representative of the exposure-outcome distributions in the overall population. Selection bias could have occurred when investigators use only one hospital data.

Answer: Thank you, we already mentioned in discussion.

Material and Methods P. 4

The inclusion criteria of the study subjects are not clear. How did you define daily counts of emergency department visits for acute respiratory diseases and cardiovascular diseases? by ICD10 or ICD10 plus some kind of treatments from hospital electronic medical records?

Answer: Thanks, we already edited this in methods part

Material and Methods P. 5

The analyses consider several important confounding variables, including age, gender, and season.

Answer: Thank you.

Discussion P.9

Although unmeasured confounding remained in this study, you provide important new evidence that short-term PM2.5, PM10 and other air pollutant exposures is significantly related to emergency department visits for acute respiratory diseases and cardiovascular diseases. Further research might be conducted to investigate the associations of long-term exposure to fine particulate matter and air pollutants with healthcare utilization.

Answer: We add this point in discussion section, thanks for suggestion.

Discussion P.10

According to previous studies, Chiang Mai has been facing severe problem of haze pollution over the past years. Did emergency department visits to treat respiratory diseases and cardiovascular diseases increase during haze episodes in your study? Had Maharaj Nakorn Chiang Mai Hospital been prepared to meet the surge in demand for medical treatment? It could be one of the confounding factors in your study and may produce spurious or distorted associations between exposure to air pollutants and increased emergency department visits. So, the authors may add methods to decrease the impact of confounding variables in the study design. This requires clarification.

Answer: Our ED has a plan for ED overcrowding, since haze pollution occur this does not impact care process in our ED. However ED overcrowded may not impact the outcome of this research.

Discussion P.10

Chiang Mai is surrounded by the mountain ranges of the Thai highlands and has a tropical savanna climate with hot, wet temperatures year-round. Many studies have shown the meteorological factors could impact the PM concentrations, including the dispersion, growth, chemical production, photolysis, and deposition of PM. Besides, the dominant meteorological factors for PM concentrations are closely related to geographical conditions. Also, weather conditions could trigger respiratory symptoms in patients with respiratory diseases. Thus, I suggest that the effects of PM concentrations on multiple meteorological factors, including cloud cover, temperature, wind, humidity, relative humidity, pressure, precipitation, rainfall, radiation and planetary boundary layer height might be included and carefully examined in the further study.

Answer: Thank you for your suggestion, we add this in limitation of revised manuscript.

Conclusion P.10

The authors should be cautious in interpreting your results. Many studies focusing on PM2.5 and health outcomes may overstate conclusions of their findings.

Reviewer #2: This is an important research study into the acute effects of ambient air pollutants on ED visits for a variety of impotant outcomes. However, there are several limitations that either make me doubt the results and/or formations of the analysis that are unclear and incomplete. Major and minor comments are provided below.

Major Comments:

1. The exposure assessment metrics are not clearly stated in the abstract or in the methods. How were pollutants measured? What is the LOD and how were below LOD values addressed? So2 cannot be 0ug/m3, which makes me concerned about how below LOD values were handled.

Answer: Sorry about the mistakes, we check this with SHIRU and found that this data was AQI not concentration, we have change to AQI not concentration.

2. It is unclear why PM2.5 was only modeled as 50ug/m3 as the reference group. What is the rationnale for this? Why not model a continuous measure? This analysis seems quite incomplete to me.

Answer: 50 (AQI) is the upper limit of good level of AQI, from our reference.

3. PM10 was substantially lower than PM2.5. This practically speaking cannot be possible because PM2.5 is a fraction of PM10. Authors need to explain this oddity. Was PM2.5 and PM10 somehow swapped? This is a major gap in the data that is not addressed by the authors.

Answer: From the mistake that we mention earlier, we check this with SHIRU and found that this data was AQI not concentration, we have change to AQI not concentration.

4. There is a lot of ambiguities that need clearing up, which I believe is largely due to issues with the grammar. For instance, referring to exposure on the following day in the abstract, and repeating it later, makes it sound like exposure effects on the day after ED visit was associated with ED visits. This is inconsistent with how other day lags are talked about in the paper.

Answer: Thank you for your suggestion. We edit our manuscript many points.

5. Overall, there needs to be much more extensive english language editing to fix the clarity of this manuscript.

Answer: In revised version, we edit by English language specialist.

6. Study figures, especially the last two, are of very poor quality and mostly unreadable. These must be more clear for the reader to understand the data.

Answer: Our picture was more than 900 DPI, in our opinion, the publisher was reduce resolution of before send to the reviewer.

Minor Comments:

1. The abstact background is lacking background information.

Answer: we just added this in formation in abstract.

2. the abstract methods need to specify the study area (city, country) and the exposure assessment method

Answer: Thank you, we just added this in formation in abstract.

3. What statistical package (and version number) did authors use for dlnm modeling? Need to specify these

Answer: Thank you, we just added this in formation in abstract.

6. PLOS authors have the option to publish the peer review history of their article (what does this mean?). If published, this will include your full peer review and any attached files.

Do you want your identity to be public for this peer review? For information about this choice, including consent withdrawal, please see our Privacy Policy.

Reviewer #1: Yes: Man-Ju Ting

Reviewer #2: Yes: Eric S Coker

---

## [Decision Letter · Decision Letter 1]

5 Feb 2023

PONE-D-22-17550R1Association Between Air Quality Index and Effect of Emergency Department Visits for Acute Respiratory Disease, Acute Coronary Syndrome, Acute Heart Failure, and StrokePLOS ONE

Dear Dr. Wittayachmnankul,

Thank you for submitting your manuscript to PLOS ONE. After careful consideration, we feel that it has merit but does not fully meet PLOS ONE’s publication criteria as it currently stands. Therefore, we invite you to submit a revised version of the manuscript that addresses the points raised during the review process.

We look forward to receiving your revised manuscript.

Kind regards,

Oyelola A. Adegboye, PhD

Academic Editor

PLOS ONE

Reviewers' comments:

Reviewer's Responses to Questions

**Comments to the Author**

1. If the authors have adequately addressed your comments raised in a previous round of review and you feel that this manuscript is now acceptable for publication, you may indicate that here to bypass the “Comments to the Author” section, enter your conflict of interest statement in the “Confidential to Editor” section, and submit your "Accept" recommendation.

Reviewer #1: All comments have been addressed

Reviewer #3: (No Response)

2. Is the manuscript technically sound, and do the data support the conclusions?

Reviewer #1: Partly

Reviewer #3: Partly

3. Has the statistical analysis been performed appropriately and rigorously? 

Reviewer #1: Yes

Reviewer #3: Yes

4. Have the authors made all data underlying the findings in their manuscript fully available?

Reviewer #1: Yes

Reviewer #3: Yes

5. Is the manuscript presented in an intelligible fashion and written in standard English?

Reviewer #1: No

Reviewer #3: Yes

6. Review Comments to the Author

Reviewer #1: Dear editors and authors,

Thank you for your thoughtful revisions to the manuscript. Authors have revised some issues of concern that I outlined and have stated the study limitations that may affect the interpretation of the results in a balanced tone. I appreciate the effort the authors put into this work. The article is clearer, and the research question is interesting. However, there are still several points should be addressed before this manuscript can be suitable for publication.

1. Editing for the English language is still required. Please revise and perfect your manuscript by a professional language polishing service or a native English speaker and attach the language editing certificate when resubmitting the manuscript.

2. Title, page 1, line 1-2.

The title of your manuscript has changed to “Association Between Air Quality Index and Effect of Emergency Department Visits for Acute Respiratory Disease, Acute Coronary Syndrome, Acute Heart Failure, and Stroke.” It's probably not written in standard English, so the authors may discuss with a native English speaker about that.

3. Abstract, page 2, line 12.

It is not appropriate to add website address in the abstract section. “(https://aqicn.org/map/chiang-mai/)” could be eliminated and should be included in the reference section.

Reviewer #3: This is an important study that investigated the association between the air quality index outdoors and emergency visits for various health outcomes. These findings may contribute to policy change to improve outdoor air quality in Thailand in the future. The revision seemed to have accommodated comments from the reviewers. Additional comment is provided below.

The air quality data is collected outdoors while the location (indoor or outdoor) of the patient when the patient was transferred to ER is not clear. Most likely, the air indoors will not reflect the air outdoors due to building infiltration, filtration in HVAC systems, or the use of portable air cleaners. Can rerun your analysis by separating patients by their location when sent to ER? If this is not doable, can you explain why the air quality index outdoors can be used to hypothesize outdoor air pollution may be the cause of the ER visit?

7. PLOS authors have the option to publish the peer review history of their article (what does this mean?). If published, this will include your full peer review and any attached files.

Reviewer #1: **Yes: **Man-Ju Ting

Reviewer #3: **Yes: **Kazukiyo Kumagai

---

## [Author Response · Author response to Decision Letter 1]

16 Feb 2023

Dear Reviewer,

Thank you for your valuable feedback on our manuscript titled "Association Between Air Quality Index and Effects on Emergency Department Visits for Acute Respiratory and Cardiovascular Diseases." We appreciate the time and effort you put into reviewing our work and providing constructive comments that have helped us improve the manuscript. We have addressed all your concerns in the revised manuscript, and we hope that you find the revised version to be suitable for publication.

We have made significant changes to the manuscript based on your suggestions, including revising the title, removing inappropriate content from the abstract, and having the manuscript professionally edited by a native English speaker. Additionally, we have addressed your concern about the location of patients when they presented to the emergency department (ED) and clarified that all air quality data used in our study were collected from outdoor sources.

As per your request, we have provided some previous articles that establish the relationship between outdoor air pollution and negative health outcomes. These articles are now included in the revised manuscript.

Once again, we appreciate your thoughtful and thorough review of our work. Your feedback has been invaluable to the development of our study, and we hope that the revised manuscript meets your expectations.

Sincerely,

BORWON WITTAYACHAMNANKUL, MD, PhD.

---

## [Decision Letter · Decision Letter 2]

4 May 2023

PONE-D-22-17550R2Association Between Air Quality Index and Effects on Emergency Department Visits for Acute Respiratory and Cardiovascular DiseasesPLOS ONE

Dear Dr. Wittayachmnankul,

Thank you for submitting your manuscript to PLOS ONE. After careful consideration, we feel that it has merit but does not fully meet PLOS ONE’s publication criteria as it currently stands. Therefore, we invite you to submit a revised version of the manuscript that addresses the points raised during the review process.

We look forward to receiving your revised manuscript.

Kind regards,

Oyelola A. Adegboye, PhD

Academic Editor

PLOS ONE

Reviewers' comments:

Reviewer's Responses to Questions

**Comments to the Author**

1. If the authors have adequately addressed your comments raised in a previous round of review and you feel that this manuscript is now acceptable for publication, you may indicate that here to bypass the “Comments to the Author” section, enter your conflict of interest statement in the “Confidential to Editor” section, and submit your "Accept" recommendation.

Reviewer #1: All comments have been addressed

Reviewer #4: (No Response)

2. Is the manuscript technically sound, and do the data support the conclusions?

Reviewer #1: Yes

Reviewer #4: Yes

3. Has the statistical analysis been performed appropriately and rigorously? 

Reviewer #1: Yes

Reviewer #4: Yes

4. Have the authors made all data underlying the findings in their manuscript fully available?

Reviewer #1: Yes

Reviewer #4: (No Response)

5. Is the manuscript presented in an intelligible fashion and written in standard English?

Reviewer #1: Yes

Reviewer #4: No

6. Review Comments to the Author

Reviewer #1: Dear authors,

Thank you for your thoughtful revisions to the manuscript. Authors have revised the issues of concern that I outlined and have stated the study limitations that affect the interpretation of the results. I appreciate the effort the authors put into this work. The article is clearer, and this research is repeatable due to enough information you provide. I think your paper could be accepted for publication into PLOS ONE now.

Reviewer #4: First of all, I would like to congratulate the authors for undertaking this research as it would add to the existing body of literature in the South and South-east Asian region on the topic of air quality and its impacts on health.

Major Comments:

• In general, the authors need to get the English corrected (sentence format) either by sending it to another colleague better in English or a native English language speaker. The authors can also use various freely available websites that can be used for English editing.

• There is a major confusion, as to what is the variable used by the authors for undertaking the study, PM2.5 concentration, AQI derived from PM2.5 or AQI derived from PM2.5/PM10 or AQI derived from a calculation using concentrations of PM2.5, PM10, ozone (O3), nitrogen dioxide (NO2), and sulfur dioxide (SO2).

• Are there any adjustments done for any other physical / physiological aspects of the patients concerned for performing this study

Minor Comments:

Abstract:

1. Pg 2: Result part of the abstract, “AQI of PM2.5 concentration was 89.0 ± 40.2”, this section is creating confusion if the value is a concentration or an index. Because concentration should be followed by a unit, but AQI could just be a value. The authors can mention in the beginning that AQI was derived from PM2.5 concentrations and then follow with only one terminology that is AQI. This would reduce a lot of the confusion.

2. In the abstract itself the authors can indicate if the association was positive or negative for the purpose of clarity.

3. Could the mortality association be termed as pre-mature mortality. As this might not be the direct cause / only established cause.

Introduction

4. Pg 4: Please modify the following sentences “Air pollution is a major public health issue worldwide and represents the largest environmental problem, especially with harmful health effects.” As “Air pollution is a major public health concern worldwide and represents one of the largest environmental problems. 1”. Kindly check for these errors in the entire manuscript.

5. Pg 4: There are vague sentences that need significant modification i.e. “Thailand is a SEA country with exposure to high annual mean WHO Air Quality Guidelines (AQG).11,12”

6. Kindly refrain from using such sentences “However, no previous study in SEA mentions the effects of PM2.5 on health impacts in ED visits”. There are quiet a few studies “IJERPH | Free Full-Text | The Effects of PM2.5 from Asian Dust Storms on Emergency Room Visits for Cardiovascular and Respiratory Diseases (mdpi.com) ; Short-term PM2.5 exposure and emergency hospital admissions for mental disease - ScienceDirect”

Methodology

7. Pg 5: The authors indicate “Daily outdoor AQI of PM2.5, PM10, ozone (O3), nitrogen dioxide (NO2), and sulfur dioxide (SO2)”. So how is the AQI calculated and what has been used. There is a strong confusion in terms of the AQI terminology used in Abstract and in the text.

8. Pg 6: Again here, the authors indicate “The primary exposure variable was PM2.5”

9. Pg 6: Another statement “Xt-q represented the focused AQI of air pollutants (PM2.5 or PM10) concentration”. That means were PM10 or PM2.5 concentrations used continuously for this study or randomly any particulate concentration was used.

10. Following the four statements (three above and one in abstract), I am now in a confusion, what is the variable used by the authors for undertaking the study PM2.5 concentration, AQI derived from PM2.5 or AQI derived from PM2.5 or PM10 or AQI derived from a calculation using concentrations of PM2.5, PM10, ozone (O3), nitrogen dioxide (NO2), and sulfur dioxide (SO2).

Results and discussion

11. Pg7: 1st paragraph, the percentage of different cause specific respiratory diseases, is not coming up to 100%, kindly provide the breakup accordingly.

12. Confusing sentence “The AQI of all air pollutants was highest during April of both years (2018 and 2019), as shown in Supplementary Figure 2”. The current study period has only one April so why are the authors discussing about two Aprils.

Discussions

13. Please mention in complete sentences “This study found that AQI of PM2.5 was associated with the number of ED visits due”. What is the association?

Conclusions

14. Conclusion should be elaborated and given point wise for all the associations.

7. PLOS authors have the option to publish the peer review history of their article (what does this mean?). If published, this will include your full peer review and any attached files.

Reviewer #1: **Yes: **Man-Ju Ting

Reviewer #4: No

---

## [Author Response · Author response to Decision Letter 2]

11 Jun 2023

Dear Editor-in-Chief,

We hope this letter finds you well. We would like to submit the revised version of our manuscript entitled "Association Between Air Quality Index and Effects on Emergency Department Visits for Acute Respiratory and Cardiovascular Diseases" for consideration of publication in PLOS ONE.

We would like to express our gratitude to the reviewers and the editorial team for their valuable feedback and insightful comments on our initial submission. We have carefully addressed all the concerns raised and made significant revisions to improve the clarity, accuracy, and overall quality of the manuscript.

In response to the reviewers' comments, we have made the following major revisions:

1. Clarified the calculation of the Air Quality Index (AQI) and its usage throughout the study, ensuring consistency in terminology.

2. Provided a detailed breakdown of the respiratory diseases leading to ED visits, including percentages for clarity and completeness.

3. Elaborated on the associations observed in the study, specifying the increased risks of ED visits for Acute Coronary Syndrome (ACS) and respiratory diseases, particularly pneumonia, with relevant lag times.

4. Enhanced the conclusion section by summarizing the associations point-wise and emphasizing the need for effective measures to reduce PM2.5 pollution levels and protect vulnerable populations.

We believe that these revisions have significantly strengthened the manuscript and addressed all the concerns raised by the reviewers. We are confident that the revised version makes a valuable contribution to the scientific literature on the health effects of air pollution.

Please find attached the revised manuscript and the response to reviewer comments document, which provides a detailed point-by-point response to each reviewer's comments and outlines the changes made in the revised version.

We sincerely hope that you will find our revised manuscript suitable for publication in PLOS ONE. We believe that our findings have important implications for public health and contribute to the existing knowledge on the association between air quality and its effects on acute respiratory and cardiovascular diseases.

Thank you for considering our revised submission. We look forward to your positive response and the opportunity to contribute to the scientific discourse in your esteemed journal.

Yours sincerely,

Borwon Wittayachamnankul, MD, PhD

Department of Emergency Medicine, Faculty of Medicine, Chiang Mai University 

Chiang Mai 50200, Thailand

---

## [Decision Letter · Decision Letter 3]

21 Jul 2023

PONE-D-22-17550R3Association Between Air Quality Index and Effects on Emergency Department Visits for Acute Respiratory and Cardiovascular DiseasesPLOS ONE

Dear Dr. Wittayachmnankul,

Thank you for submitting your manuscript to PLOS ONE. After careful consideration, we feel that it has merit but does not fully meet PLOS ONE’s publication criteria as it currently stands. Therefore, we invite you to submit a revised version of the manuscript that addresses the points raised during the review process.

We look forward to receiving your revised manuscript.

Kind regards,

Oyelola A. Adegboye, PhD

Academic Editor

PLOS ONE

Reviewers' comments:

Reviewer's Responses to Questions

**Comments to the Author**

1. If the authors have adequately addressed your comments raised in a previous round of review and you feel that this manuscript is now acceptable for publication, you may indicate that here to bypass the “Comments to the Author” section, enter your conflict of interest statement in the “Confidential to Editor” section, and submit your "Accept" recommendation.

Reviewer #4: (No Response)

2. Is the manuscript technically sound, and do the data support the conclusions?

Reviewer #4: Partly

3. Has the statistical analysis been performed appropriately and rigorously? 

Reviewer #4: Yes

4. Have the authors made all data underlying the findings in their manuscript fully available?

Reviewer #4: (No Response)

5. Is the manuscript presented in an intelligible fashion and written in standard English?

Reviewer #4: Yes

6. Review Comments to the Author

Reviewer #4: It is nice to see the revised version of the manuscript. While most of the comments are addressed well but one fundamental question needs more modification that the authors are not able to explain in a logical manner. Therefore, suggesting a major revision.

The response to the 3 different comments for the use of AQI are still not clear and creating confusion.

1. The terminology that should be used throughout the manuscript is only “AQI”

2. AQI is a calculation that uses values of all pollutant parameters to reach at one meaningful value. This is termed as AQI. Have the authors done it? Please provide the equation how AQI was calculated in this manuscript.

3. Refrain from mentioning “AQI of PM2.5 or AQI of PM10 because if that is the case (separately calculated) then why not directly use concentration.

4. I am still confused, has the study used AQI of PM2.5 for further deep diving? Or it is AQI of PM10, AQI of NO2, AQI of O3 etc etc. separately or only AQI derived from calculation involving all pollutants together and not separately?

7. PLOS authors have the option to publish the peer review history of their article (what does this mean?). If published, this will include your full peer review and any attached files.

Reviewer #4: No

---

## [Author Response · Author response to Decision Letter 3]

27 Jul 2023

Response to reviewer

We appreciate the editor's and the reviewers' insightful comments. The manuscript has already undergone revisions, and we are hopeful that the updated version satisfies the requirements for publication. Changes were tracks change in the revised manuscript.

Reviewer #4: It is nice to see the revised version of the manuscript. While most of the comments are addressed well but one fundamental question needs more modification that the authors are not able to explain in a logical manner. Therefore, suggesting a major revision.

The response to the 3 different comments for the use of AQI are still not clear and creating confusion.

1. The terminology that should be used throughout the manuscript is only “AQI”

Answer: Terminology: We apologize for any confusion caused by the inconsistency in AQI terminology used throughout the manuscript. We acknowledge the importance of using a standardized term consistently, and we will revise the manuscript to ensure that only "AQI" is used consistently in all sections.

2. AQI is a calculation that uses values of all pollutant parameters to reach at one meaningful value. This is termed as AQI. Have the authors done it? Please provide the equation how AQI was calculated in this manuscript.

Answer: AQI Calculation: We sincerely apologize for the oversight in not providing the equation for AQI calculation in the initial submission. In the revised version, we will include a detailed description of the AQI calculation method used in this study. The AQI was calculated based on the concentrations of PM2.5, PM10, ozone (O3), nitrogen dioxide (NO2), and sulfur dioxide (SO2). The precise formula used to calculate the AQI will be described in the supplementary section, which is displayed below this.

3. Refrain from mentioning “AQI of PM2.5 or AQI of PM10 because if that is the case (separately calculated) then why not directly use concentration

Answer: We have verified that the source of Smoke Haze Integrated Research Unit (SHIRU) and the Air Pollution in Chiang Mai: Real-time Air Quality Index Visual Map website came form Thailand's air and noise pollution, the Pollution Control Department under the Ministry of Natural Resources and Environment at "aqmthai.com," provides only AQI values for each individual pollutant such as PM2.5, PM10, O3, NO2 and SO2. To eliminate any confusion, we will accurately state in the revised manuscript that our study used AQI values for PM2.5, PM10, O3, NO2, and SO2 separately.

4. I am still confused, has the study used AQI of PM2.5 for further deep diving? Or it is AQI of PM10, AQI of NO2, AQI of O3 etc etc. separately or only AQI derived from calculation involving all pollutants together and not separately?

Answer: In our study, we used the Air Quality Index (AQI) values derived from individual pollutant concentrations to assess their associations with health outcomes. Specifically, we obtained separate AQI values for PM2.5, PM10, O3, NO2 and SO2.

---

## [Editor Report · Decision Letter 4]

2 Oct 2023

PONE-D-22-17550R4Association Between Air Quality Index and Effects on Emergency Department Visits for Acute Respiratory and Cardiovascular DiseasesPLOS ONE

Dear Dr. Wittayachmnankul,

Thank you for submitting your manuscript to PLOS ONE. After careful consideration, we feel that it has merit but does not fully meet PLOS ONE’s publication criteria as it currently stands. Therefore, we invite you to submit a revised version of the manuscript that addresses the points raised during the review process.

We look forward to receiving your revised manuscript.

Kind regards,

Oyelola A. Adegboye, PhD

Academic Editor

PLOS ONE

Journal Requirements:

Additional Editor Comments (if provided):

Editor: I am happy to consider your manuscript for publication after the following revision.

1. Separate the Materials and Methods to "Data Collection" and "Data Analysis"

2. The DLNM equation is confusing. You have RH, temperature and AP, but the other variables were X_t. Why not write them to be consistent?
---

## [Author Response · Author response to Decision Letter 4]

25 Oct 2023

Dear Editor,

I sincerely appreciate your consideration of my manuscript and the opportunity for revision.

Regarding your valuable feedback, I will make the following revisions:

 Separate the Materials and Methods to "Data Collection" and "Data Analysis"

Answer: We separate the "Materials and Methods" section into two distinct sections, namely "Data Collection" and "Data Analysis”.

 The DLNM equation is confusing. You have RH, temperature and AP, but the other variables were X_t. Why not write them to be consistent?

Answer: understand your concern about the inconsistency in variable notation within the DLNM equation. To address this, I will ensure that all variables are consistently represented as "X_t" throughout the equation, which will enhance the overall clarity and coherence of the mathematical formulation as revise as 

‘log⁡(E(Y_t ))=α+ns(〖RH〗_t,3)+ns(Temperature,3)+ns(〖Focused_AP〗_t,3)+∑_(i=0)^q▒〖β_i 〖(Focused_AP)〗_(t-i) 〗+ε_t

Where Focused_AP∈{AQI of PM_2.5,AQI of PM_10} and 

 Other_AP∈{AQI of PM_2.5,AQI of PM_10,AQI of O_3,AQI of 〖NO〗_2 }-{Focused_AP}’

Thank you for your guidance and the opportunity to improve the manuscript. I will diligently work on these revisions and resubmit the manuscript accordingly.

Best regards,

Borwon Wittayachamnankul, MD, PhD

Department of Emergency Medicine, Faculty of Medicine, Chiang Mai University 

Chiang Mai 50200, Thailand

---

## [Editor Report · Decision Letter 5]

26 Oct 2023

Association Between Air Quality Index and Effects on Emergency Department Visits for Acute Respiratory and Cardiovascular Diseases

PONE-D-22-17550R5

Dear Dr. Wittayachmnankul,

We’re pleased to inform you that your manuscript has been judged scientifically suitable for publication and will be formally accepted for publication once it meets all outstanding technical requirements.

Kind regards,

Oyelola A. Adegboye, PhD

Academic Editor

PLOS ONE
---

## [Editor Report · Acceptance letter]

9 Nov 2023

PONE-D-22-17550R5 

Association Between Air Quality Index and Effects on Emergency Department Visits for Acute Respiratory and Cardiovascular Diseases 

Dear Dr. Wittayachamnankul:

I'm pleased to inform you that your manuscript has been deemed suitable for publication in PLOS ONE. Congratulations! Your manuscript is now with our production department. 

Kind regards, 

on behalf of

Assoc Prof Oyelola A. Adegboye 

Academic Editor

PLOS ONE